# Enhancing AI Capabilities on the Abstraction and Reasoning Corpus: A Path Toward Broad Generalization in Intelligence

## Abstract

This position paper explores advancing artificial intelligence by improving its ability to generalize beyond training data, a key requirement for tasks in the Abstraction and Reasoning Corpus (ARC). Inspired by historical algorithmic challenges like the Bongard Problems, ARC tasks require pattern recognition and logical reasoning, pushing AI toward more flexible, human-like intelligence. We investigate DreamCoder, a neural-symbolic system, and the role of large language models in ARC. We emphasize the need for diverse data sources, inspired by human trials and synthetic data augmentation, and propose pipelines for logical reasoning using math-inspired neural architectures. This work underlines how ARC can guide AI research, bridging the gap between machine learning and mathematical discovery.

## Introduction

Current AI systems excel in highly specific tasks but often fail to generalize to new, untrained scenarios. The limitations become evident when AI systems encounter unexpected inputs, highlighting the need for models capable of robust reasoning and adaptation. The Abstraction and Reasoning Corpus (ARC), a set of logic-based visual puzzles, serves as a rigorous benchmark to evaluate an AI's ability to solve tasks that require abstraction and reasoning (Chollet 2019). Drawing inspiration from classic challenges like the Bongard Problems, the ARC benchmark pushes AI systems beyond traditional pattern recognition towards "broad generalization" (Chollet 2019).

This position paper suggests approaches aimed at bridging this gap, from structured, symbolic programming methods to large-scale neural models, and considers how insights from human problem-solving can inform AI development. Furthermore, we discuss the importance of new data sources and the role of mathematical insights in developing systems capable of human-like logical reasoning.

Human intelligence stands out for its adaptability and ability to reason abstractly, qualities that remain elusive in AI. The Bongard Problems, introduced in 1967, were among the first tests of machine intelligence, challenging systems to discern abstract relationships in visual patterns. Although AI has made significant strides, no system has yet demonstrated the capability to solve these problems without human-designed heuristics. Similarly, ARC presents logic

puzzles that involve recognizing transformations in colored grids, aiming to test a machine's capacity for broad generalization beyond examples it has encountered in the training set. Despite advances, AI solutions to ARC have relied on predefined rules and struggle to address the broader generalization that the benchmark demands.

## Current Approaches and Challenges

Existing AI models primarily rely on large datasets and pattern recognition. Recently two complementary approaches have been used to tackle ARC tasks. One prominent approach involves DreamCoder, a neurosymbolic system that employs neural networks to construct programs that solve tasks through transformations of primitives (Alford 2021; Banburski et al. 2020; Bober-Irizar and Banerjee 2024). By breaking down problems into functional steps, DreamCoder narrows down the search space, simulating a form of reasoning.

Another approach involves Large Language Models (LLMs) such as GPT-4, which demonstrate emergent abilities across various tasks by leveraging vast amounts of training data (Bober-Irizar and Banerjee 2024; Bubeck et al. 2023). These models perform surprisingly well on ARC when tasks are translated into text, but still face limitations in precise, logic-based reasoning.

These findings reveal both the strengths and limitations of each approach. DreamCoder performed well in on ARC, demonstrating the value of symbolic reasoning systems in tasks that demand a high level of abstraction (Bober-Irizar and Banerjee 2024). LLMs, while impressive in their breadth of training data, were constrained in certain tasks but performed better on others, suggesting that LLMs and symbolic systems may be complementary in tackling ARC (Bober-Irizar and Banerjee 2024).

We advocate for a hybrid approach that leverages both neural-symbolic techniques and the generalized capabilities of foundation models.

## Adding Data to Enhance AI Performance on ARC

In addition to algorithmic innovations, more data is essential for advancing AI's abstraction and reasoning abilities on tasks like those in ARC. Data can be gathered through

controlled trials with human participants, observing how they approach and solve ARC problems (Acquaviva et al. 2021; Peterson et al. 2021). Such studies can reveal cognitive strategies that might inspire new AI techniques.

Another valuable source of data is augmentation or synthetic data generation, creating a richer and more extensive dataset for model training.

This situation parallels early AI efforts in games such as Go, where models initially struggled to generalize. The breakthrough came with AlphaGo, which "solved" Go by training on vast amounts of human-played games, leveraging both real and augmented data to develop advanced strategies. Similarly, AI systems tackling ARC could benefit from massive datasets, potentially incorporating data on human reasoning patterns or large-scale synthetic examples to mimic abstract problem-solving. Expanding data sources in this way could bridge some gaps in ARC and improve AI's capacity for reasoning.

## Building Pipelines for Human-Like Logical Reasoning with Neural Networks in ARC

The ARC challenge highlights a critical gap in AI: the need for human-like reasoning and abstraction, which remains largely unmet by current models. To tackle ARC tasks, a promising strategy involves pipelines that combine neural network flexibility with the rigour of mathematical logic. This approach aligns with the goals of AI-assisted mathematical discovery in several ways:

1. Mathematics-Inspired Neural Architectures:

   ARC tasks require precise transformations and logical rules to generalize patterns from limited examples: a skill neural networks currently lack. By incorporating mathematical principles into model design, we could build architectures that better emulate logical rigour, allowing networks to generate solutions with human-like reasoning processes. For example, category theory, symbolic logic, or topology could inspire models with structured, interpretable layers that manage abstract concepts as ARC demands. Such architectures could more effectively learn the rules underlying ARC's transformations, yielding a step closer to human-level generalization.

2. Neural Networks in Mathematical Discovery and ARC:

   Just as neural networks can aid in uncovering new insights in mathematics, they could be applied to ARC to discover novel solution patterns or paradigms. By training on ARC tasks, networks may reveal new ways of representing abstract reasoning that humans have not yet explored. This process can contribute to both AI and mathematical research by identifying overlooked structures or insights, potentially offering fresh approaches to abstraction and reasoning.

This dual approach — using mathematics to inspire model architectures and leveraging neural networks to advance mathematical understanding—could push AI closer to achieving the abstract and adaptable reasoning abilities that ARC challenges. This would make ARC a testing ground for both AI robustness and mathematical discovery in reasoning.

## Complementing human problem solving by collaborating with algorithms

As AI researchers continue to tackle ARC, shifting from an "AI versus human" mindset toward a "human-AI cooperation" paradigm maybe more effective. Instead of expecting AI models to autonomously solve all ARC problems, we can design frameworks where AI models work alongside human experts, each addressing different aspects of the challenge.

This cooperative approach has several advantages. For one, AI models (such as LLMs) are adept at identifying patterns, generating plausible transformations, and suggesting hypotheses based on existing data. They act as a source of inspiration or an analytical tool, providing humans with insights or suggesting strategies that may not be immediately apparent.

On the other hand, humans bring abstract reasoning, intuition, and contextual understanding to the process, which are areas where LLMs still face limitations. By combining human insight with machine-driven pattern recognition, we may be able to create a powerful team capable of solving ARC problems more efficiently.

For example, an LLM might solve certain ARC tasks through brute-force pattern matching or by generating novel transformations that are beyond human imagination, while a human collaborator refines, evaluates, or adjusts these transformations to better fit nuanced ARC requirements. This collaboration also addresses some limitations of LLMs, such as difficulties in interpreting ambiguous or complex visual patterns, which human partners can more easily resolve.

In practical terms, this hybrid approach could involve developing interactive interfaces or cooperative problem-solving pipelines where humans can interact with LLMs, query them, or refine their output in real-time. Such setups can capitalize on the LLMs' ability to explore large solution spaces and test multiple hypotheses while allowing humans to provide oversight. Ultimately, this partnership allows each party—human and machine—to contribute their unique strengths. This may lead to more robust solutions and potentially advance our understanding of broad generalization in intelligence.

## Conclusion

ARC presents an essential benchmark for developing AI systems capable of reasoning and abstraction. By combining data-driven techniques and structured mathematical principles, we can build neural architectures that support rigorous logical reasoning. Expanding ARC data through human trials and synthetic methods (inspired by AlphaGo's success with data-rich Go training), can aid generalization. Furthermore, employing math-inspired pipelines may help align neural networks with human-like logic, promoting insights applicable to both AI and mathematical science. Such advancements would mark a significant step toward robust AI that can tackle diverse, complex real-world tasks.

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
