# OpenReview forum: "Enhancing AI Capabilities on the Abstraction and Reasoning Corpus: A Path Toward Broad Generalization in Intelligence"
_AAAI.org/2025/Workshop/NeurMAD — AAAI 2025 Workshop NeurMAD Submission_

### Official Review · Reviewer_4Q3d · 2024-12-18

**Rating:** 6
**Confidence:** 3

**Review:**

This paper explores a dual approach to align AI in human-like reasoning, combining structured neural-symbolic methods with the adaptive capabilities of Large Language Models (LLMs). Focused on the Abstraction and Reasoning Corpus (ARC) as a benchmark, it highlights how these techniques can complement human problem-solving and advance abstraction and broad generalization in AI systems.

1) The paper effectively underscores a critical limitation of current AI systems: their inability to generalize and reason abstractly in human-like ways. The emphasis on ARC as a benchmark for "broad generalization" is well-motivated, presenting a relevant challenge to AI research.

2) The discussion of DreamCoder and Large Language Models (LLMs) highlights their complementary strengths. The hybrid approach proposed by leveraging neural-symbolic techniques alongside foundation models shows thoughtful integration of existing tools. In particular to overcome abstract reasoning, intuition, and contextual understanding where LLMs still struggle.

3) While the paper promotes structured architectures inspired by mathematical logic, it does not convincingly argue why such structures alone would suffice to bridge the gap between current AI capabilities and human reasoning. I would appreciate more clarity and examples on the mathematical structures and tools that the authors claim should be used.

---

### Official Review · Reviewer_8ScV · 2024-12-20
**The paper presents intriguing ideas for advancing AI generalization through ARC but suffers from a lack of empirical evidence, speculative analogies, impractical collaboration details, and insufficient focus on ARC-specific challenges, leaving its proposals largely ungrounded.**

**Rating:** 3
**Confidence:** 3

**Review:**

The paper explores advancing artificial intelligence by addressing its limitations in generalization through the Abstraction and Reasoning Corpus (ARC), a benchmark for logic-based tasks requiring human-like reasoning. It critiques the narrow capabilities of current AI models, highlighting the neurosymbolic DreamCoder system for its structured reasoning and LLMs like GPT-4 for their adaptability while advocating for hybrid approaches that combine their strengths. The authors propose enriching AI capabilities through data augmentation, including synthetic datasets and human trial observations, and developing math-inspired neural architectures that embed logical rigor. They also emphasize human-AI collaboration, suggesting interactive frameworks where humans and machines jointly solve ARC tasks by leveraging complementary strengths.

Key points to address:
1. The paper heavily theorizes without presenting empirical results or concrete benchmarks for its proposed hybrid models or math-inspired architectures. This omission weakens the argument for their efficacy.
2. Drawing analogies with AlphaGo's approach and mathematical discovery seems speculative without real evidence that these strategies would generalize to ARC, which differs fundamentally from Go in structure and problem-solving requirements.
3. While the proposal for human-AI collaboration is cool, it lacks practical implementation details, such as how interactive interfaces would function or how the collaboration pipeline would be evaluated.
4. The paper overlooks the unique challenges of ARC, such as its focus on abstract transformations that defy straightforward data-driven solutions. This weakens its proposals for data augmentation and math-inspired architectures, which may not align with ARC’s core demands.
5. The paper blends philosophical aspirations of AI generalization with technical proposals without a clear roadmap for achieving its goals.

---

### Official Review · Reviewer_kmZS · 2024-12-24
**Some interesting high-level ideas but limited novelty**

**Rating:** 4
**Confidence:** 4

**Review:**

Summary:
- This position paper investigates the role of language models and neurosymbolic systems for solving ARC. It proposes to use a hybrid approach, combining the strenghts of LLMs and math-inspired neural architectures.
Further, it suggests using data augmentation in order to advance the abstraction and reasoning abilities of neural models.

Strengths:
- Summary of current approaches at solving ARC
- Advocation for a combination of neural networks with the rigor of mathematical logic

Weaknesses:
- While this position paper includes a brief summary of current and potential future approaches of solving ARC, it does not really provide any novel insights (compare (Bober-Irizar and Banerjee 2024)).
- Human collaboration:  On one hand this is not the point of ARC (see Chollet 2019). On the other hand, the potential for improvement would be quite small, as humans are already very good at solving ARC (LeGris, Solim, et al. "H-ARC: A Robust Estimate of Human Performance on the Abstraction and Reasoning Corpus Benchmark." arXiv preprint arXiv:2409.01374 (2024)).
- Data augmentation: While data augmentation might certainly help with ARC, it is unclear if it actually helps neural models in achieving the generalization required to solve ARC, or if it only tries to move the test-tasks in-distribution.
- The idea of combining neural architectures with math-inspired principles is interesting, however not novel.
(Wang, Ruocheng, et al. "Hypothesis search: Inductive reasoning with language models." arXiv preprint arXiv:2309.05660 (2023)) and (Barke, Shraddha, et al. "HYSYNTH: Context-Free LLM Approximation for Guiding Program Synthesis." The Thirty-eight Conference on Neural Information Processing Systems (2024)) and (Kalyanpur, Aditya, et al. "Llm-arc: Enhancing llms with an automated reasoning critic." arXiv preprint arXiv:2406.17663 (2024)) have already proposed and evaluated similar ideas.
- Deep-learning guided program synthesis has been used quite extensively in the latest ARC Prize 2024, so the ideas have been known for a while (Chollet, Francois, et al. "ARC Prize 2024: Technical Report." arXiv preprint arXiv:2412.04604 (2024).)

Relevant papers not properly cited, e.g.
 - Bongard Problems (M. Bongard. Pattern Recognition. Spartan Books, New York, 1970.)
 - AlphaGo (Silver, David, et al. "Mastering the game of Go with deep neural networks and tree search." nature 529.7587 (2016): 484-489.)
 - DreamCoder (Ellis, Kevin, et al. "Dreamcoder: Bootstrapping inductive program synthesis with wake-sleep library learning." Proceedings of the 42nd acm sigplan international conference on programming language design and implementation. 2021.)
 - Early works on using language/LLMs for solving ARC, e.g. (Camposampiero, Giacomo, et al. "Abstract visual reasoning enabled by language." Proceedings of the IEEE/CVF Conference on Computer Vision and Pattern Recognition. 2023.) and (Acquaviva, Sam, et al. "Communicating natural programs to humans and machines." Advances in Neural Information Processing Systems 35 (2022): 3731-3743.)

Overall, many of the points brought up in this position paper follow the work by (Bober-Irizar and Banerjee 2024), with limited novelty in its ideas.

---

### Decision · Program_Chairs · 2024-12-30

**Decision:**

Reject

**Comment:**

 We agree with the major opinions of  the reviewers.